# LUM-ViT: Learnable Under-sampling Mask Vision Transformer for Bandwidth Limited Optical Signal Acquisition

**Lingfeng Liu[1], Dong Ni[2]\*, Hangjie Yuan[2]**
[1]College of Information Science & Electronic Engineering, Zhejiang University
[2]College of Control Science and Engineering, Zhejiang University
{llfeng,dni,hj.yuan}@zju.edu.cn

## Abstract

Bandwidth constraints during signal acquisition frequently impede real-time detection applications. Hyperspectral data is a notable example, whose vast volume compromises real-time hyperspectral detection. To tackle this hurdle, we introduce a novel approach leveraging pre-acquisition modulation to reduce the acquisition volume. This modulation process is governed by a deep learning model, utilizing prior information. Central to our approach is LUM-ViT, a Vision Transformer variant. Uniquely, LUM-ViT incorporates a learnable under-sampling mask tailored for pre-acquisition modulation. To further optimize for optical calculations, we propose a kernel-level weight binarization technique and a three-stage fine-tuning strategy. Our evaluations reveal that, by sampling a mere 10% of the original image pixels, LUM-ViT maintains the accuracy loss within 1.8% on the ImageNet classification task. The method sustains near-original accuracy when implemented on real-world optical hardware, demonstrating its practicality. Code will be available at https://github.com/MaxLLF/LUM-ViT.

## 1 Introduction

In today's data-driven world, efficient data acquisition and processing are pivotal across numerous domains. Despite breakthroughs in sensing techniques, bandwidth constraints remain a persistent hurdle in applications such as hyperspectral detection (Wang & Xiao, 2020), remote sensing (Altamimi & Ben Youssef, 2022), biological optical imaging (Park et al., 2021), ultra-high-definition imaging (Zhou et al., 2020), and high-demand detection (Mehmood & Anees, 2020), to name a few.

Hyperspectral imaging (HSI) (Ronan et al., 2022) is a pivotal example of capturing 2D images across a vast electromagnetic spectrum range with acute spectral resolution. Within the canonical acquisition-storage-transmission-processing sequence, the prodigious volume of HSI data introduces substantial challenges, notably concerning bandwidth capacities (Fowler, 2014). Upon transitioning to the processing phase, traditional methodologies, typically tailored for low-intensity data, grapple with the complexities inherent in handling thousand-channel hyperspectral information. As a remedy, these techniques frequently turn to feature extraction to reduce dimensionality (Rasti et al., 2020). Notably, unlike structured high-level data such as language, image data—being more elementary—harbors a greater degree of redundancy. This notion is echoed by He et al. (2021), suggesting that a mere subset of input tokens adequately reconstruct the corrupted view.

Despite its earlier introduction, Compressive Sensing (CS) theory (Donoho, 2006) remains a valuable reference for pre-acquisition modulation. The central tenet of CS is its ability to reconstruct original data from under-sampled representations, assuming the data is inherently sparse. Applications like single-pixel imaging (Suo et al., 2016), rooted in the CS principle, exhibit potentials across various imaging domains using a digital micromirror device (DMD) (Zhang et al., 2021). However, this method is effective only with relatively high under-sampling rates. Indeed, the reconstructed images become unrecognizable to humans when the under-sampling rate falls below 30%. A pivotal shortfall of the CS approach as a signal processing technique is its incapacity to leverage prior

---

\*Corresponding Author

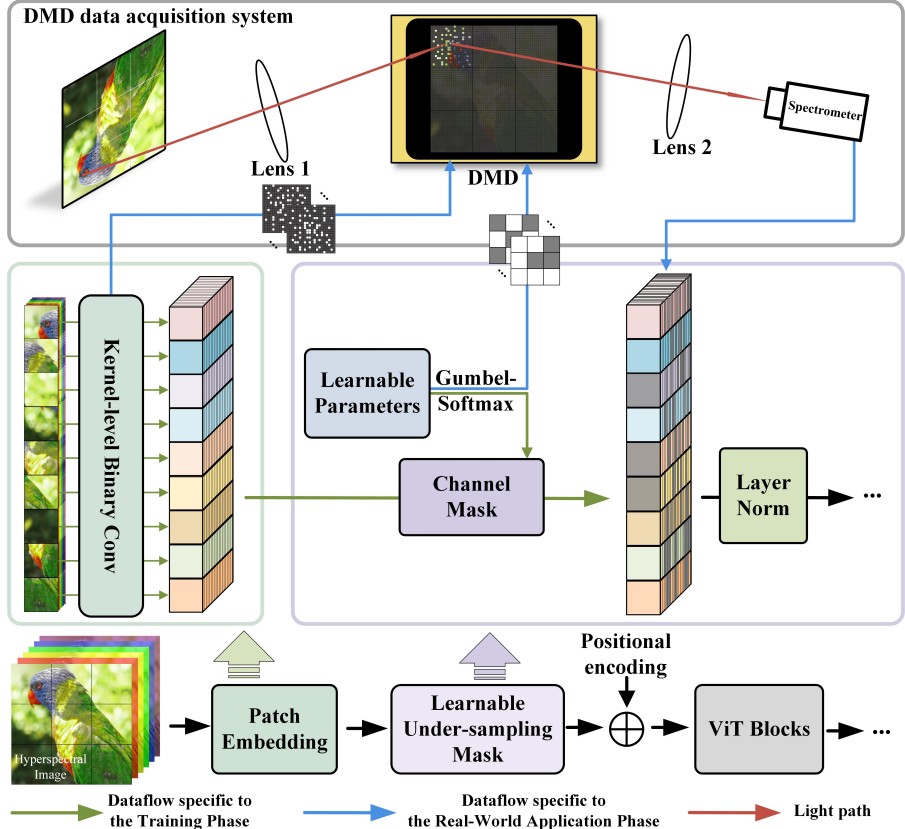

**Figure 1: The comprehensive structure of LUM-ViT.** The method unfolds in two stages: the electronic-only training and infernce phase, referred to as the Training Phase, with dataflow depicted in green, and the DMD-involved inference phase, referred to as the Real-World Application Phase, with dataflow depicted in blue. The illustration shows how the first patch is modulated by the DMD using the first kernel, a process replicable to all kernels and patches, though not depicted. The learnable under-sampling mask determines their selection.

knowledge on the detection subject. It is anticipated that methods capable of harnessing this prior information could surpass this traditional compressive sensing method.

Deep learning (DL), a burgeoning branch of machine learning, emerges as a compelling alternative for its capability to use prior knowledge when given abundant training data. Deep learning methods like Convolutional Neural Networks (CNN) (Simonyan & Zisserman, 2014), Vision Transformers (ViT) (Yuan et al., 2021), and Recurrent Neural Networks (Cho et al., 2014) stand out in feature extraction and processing for multispectral and hyperspectral data (Pang et al., 2021; Zhang et al., 2023; Lee et al., 2023; Islam et al., 2023). Hyperspectral-specific Transformer variants such as MST (Cai et al., 2022a) and MST++ (Cai et al., 2022b) have been designed for efficient hyperspectral data processing and achieved exceptional results.

Many models largely assume negligible data acquisition costs, focusing on data processing through efficient network architectures (Zhou et al., 2021; Zamir et al., 2022) and robust pre-training schemes (Xue et al., 2022). However, in hyperspectral imaging processing, data acquisition is time-consuming, with a single image acquisition taking minutes or more versus a few seconds for processing on a 2080Ti. This disparity highlights the need for a deep learning approach integrated with optical hardware. Rather than accelerating data processing, we aim to devise a method for pre-acquisition modulation, reducing data volume overhead from the beginning.

Specifically, we aim to employ DMD to perform calculations of the patch-embedding layer within the ViT model. Over recent years, DMD has been utilized as a spatial light modulator in optical neural network research, primarily executing the role of Linear layers (Wang et al., 2021; Bueno et al., 2018; Liu et al., 2023a). To our knowledge, LUM-ViT is at the forefront of adopting DMD for patch-embedding. *A central aspect of our work involves tailoring the patch-embedding layer to align with DMD operations.* The inherent function of a single DMD operation entails execut-

ing spatial binary modulation across all spectral channels, using a consistent 2-D display pattern. To accommodate this feature, we have devised a patch-embedding layer comprising a one-channel kernel-level weight binarized convolutional layer.

We obtain patch-embedding outputs through pre-acquisition optical modulation and minor post-acquisition computations, with each output point corresponding to a single DMD operation. To achieve under-sampling (reducing the required optical modulation and sampling instances), we employ a learnable mask with trainable parameters refined during training to selectively retain essential points for downstream tasks from the patch-embedding outputs. Then, only the retained points are acquired in real-world signal acquisition, while others are bypassed. Note that in a single inference (*i.e.*, detecting a single target object), the masking strategy must be determined before the target object is detected and cannot be adjusted based on yet-to-know information. In neural network pruning tasks (Wimmer et al., 2023) like DynamicViT (Rao et al., 2023) and AdaViT (Meng et al., 2021), learnable masks simplify networks while retaining performance. Unlike these works, LUM-ViT requires mask scheme determination pre-acquisition and generates a mask fixed after training. The mask is suited for the validation set and downstream tasks—through training on a compatibly distributed training set. Furthermore, lightweighting the backbone is not our objective.

Building upon the described design, we developed LUM-ViT, as illustrated in Fig 1. LUM-ViT incorporates a patch-embedding layer consisting of a kernel-level binarized single-channel convolutional layer tailored for DMD operations, along with a learnable mask for under-sampling purposes. We also devised a three-stage training strategy to train LUM-ViT effectively.

We assessed LUM-ViT on the ImageNet-1k classification task. With an under-sampling rate of less than 10%, LUM-ViT exhibited a minor accuracy degradation of 1.8%, and with rates less than 2%, the accuracy drop remained under 5.5%. We further subjected LUM-ViT to real-world tests. A 4% performance drop from software-based results due to hardware-induced error, underscored LUM-ViT's real-world feasibility. Finally, experiments with actual hyperspectral data confirm LUM-ViT's capability in processing real-world hyperspectral information.

## 2 RELATED WORK

**DMD Signal Acquisition System.** The Digital Micromirror Device (DMD) encompasses a matrix of microscopic, electronically actuated mirrors. Each micromirror can be individually manipulated, toggling between two states to steer incoming light towards designated directions. As shown in Fig.1, light from the object is directed onto the DMD by Lens 1. The DMD modulates the light spatially, which Lens 2 then concentrates onto a spectrometer. This sensor excels in discerning light intensity across diverse spectra. Utilizing its pre-acquisition light modulation facility, the DMD signal acquisition system has been employed in single-pixel imaging(Suo et al., 2016) under the purview of CS theory. For details of the DMD, please refer to Section D in the Appendix.

**Vision Transformers (ViT).** Originally developed for natural language processing tasks (Vaswani et al., 2017), transformer architectures have quickly gained traction within the computer version community. Challenging the longstanding dominance of CNNs, these models have shown significant efficacy across various visual tasks, including image classification, object detection, and semantic segmentation. ViT (Dosovitskiy et al., 2021) is noteworthy as it initiates using a purely transformer-based architecture for image classification. Subsequent developments like DeiT (Touvron et al., 2021), MAE (He et al., 2021), and Swin (Liu et al., 2021) have enhanced performance and broadened applicability across visual tasks. A distinct feature of ViT and its derivatives is the patch-embedding layer, which segments an image into fixed-size, non-overlapping patches, typically using a convolutional layer with its stride equal to its kernel size.

**Transformer Pruning.** Pruning primarily aims to streamline models by discarding less crucial elements for performance. In the context of Transformers, pruning typically focuses on patches, heads, or blocks with methods like Adaptive Sparse ViT (Liu et al., 2023b), Heat-ViT (Dong et al., 2023), and Evo-ViT (Xu et al., 2021) exemplifying this approach. Identifying which network components to prune is critical. Models such as DynamicViT (Rao et al., 2023) and AdaViT (Meng et al., 2021) address this by employing learnable masks. They utilize the Gumbel-Softmax technique to morph the intrinsically non-differentiable mask selection issues into differentiable probability scenarios, ensuring the masks remain trainable. However, these methods are unsuitable for under-sampling as under-sampling requires masks on pre-acquisition optical calculations for yet-to-capture objects.

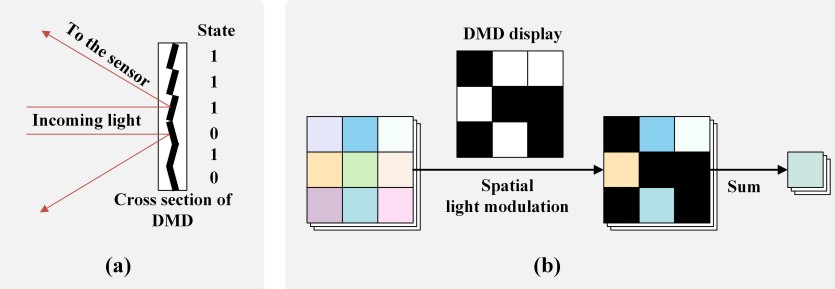

Figure 2: **The function of the DMD acquisition system.** **(a)** illustrates the spatial modulation principle of incoming light by the DMD. **(b)** outlines the modulation process. The multichannel input matrix undergoes binary masking via the DMD, following its display pattern. Lens 2 then optically sums the residual pixels.

## 3 METHOD

### 3.1 PRELIMILARIES

LUM-ViT leverages prior information to modulate signals, maintaining performance under low under-sampling conditions. Utilizing prior knowledge involves training the model on a compatibly distributed training set. Therefore, the method unfolds in two stages: Initially, we undertake training on electronic computing hardware using the training set, followed by electronic inference for evaluation. We mark this as the **Training Phase**. Subsequently, we operates DMD-involved reference to assess the real-world performance. Before data acquisition, the system is oblivious to the target object. During acquisition, the DMD optically modulates the target information prior to capture and then relays it to the electronic system for subsequent processing. We mark this as the **Real-World Application Phase**.

Fig. 2 illustrates the function of the DMD acquisition system. It's pertinent to note that within the entire system, a single DMD operation serves to execute spatial binary modulation across all spectral channels employing a consistent display pattern. We represent a single signal acquisition under one DMD operation as $\text{DMD}(\cdot, \cdot)$. Assuming a DMD display pattern as $\boldsymbol{\theta} \in \{0, 1\}^{* \times *}$, the process for a $C_h$-channel 2-D input matrix $\boldsymbol{X} \in \mathbb{R}^{* \times * \times C_h}$ with identical spatial dimensions can be articulated as:

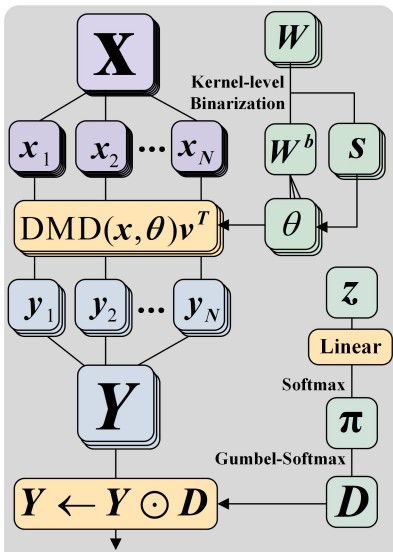

Figure 3: **The dataflow in LUM-ViT.**

$$\text{DMD}(\boldsymbol{X}, \boldsymbol{\theta}) = [\text{sum}(\boldsymbol{\theta} \odot \boldsymbol{X}_{:,:,1}), \text{sum}(\boldsymbol{\theta} \odot \boldsymbol{X}_{:,:,2}), ..., \text{sum}(\boldsymbol{\theta} \odot \boldsymbol{X}_{:,:,C_h})] \in \mathbb{R}^{C_h}, \quad (1)$$

where $\odot$ represents the Hadamard product. Across every spectral channel of $\boldsymbol{X}$, a single $\text{DMD}(\cdot, \cdot)$ spatially modulates with the same pattern, producing an intensity value for each channel.

The $\text{DMD}(\cdot, \cdot)$ operation is entirely optical. Given the inherent parallelism of optical computation, the efficiency of $\text{DMD}(\cdot, \cdot)$ remains constant and unaffected by the sizes of matrices engaged in the computation, provided they fit within the DMD's physical pixel dimensions. Therefore, *the overall efficiency is solely contingent on the number of $DMD(\cdot, \cdot)$ operations.* In scenarios with a fixed-size 2-D input image, enhancing under-sampling performance can be achieved by ensuring that each $\text{DMD}(\cdot, \cdot)$ operation covers as many pixels as possible, striving to trade the computational load in a single operation for the quantity of operations.

A straightforward approach might involve executing the DMD operation across the whole image, essentially utilizing a Linear layer. However, for standard images sized $224 \times 224$, this layer becomes overparameterized and inefficient. A more efficient strategy entails partitioning the input image into equal-sized, non-overlapping patches, then applying the $\text{DMD}(\cdot, \cdot)$ operation to each patch. The patch-embedding technique aligns well with the requirements of this strategy.

Since the DMD can only perform 2-D spatial optical modulation, it necessitates single-channel convolution for the patch-embedding layer. Specifically, for a given image $\boldsymbol{X} \in \mathbb{R}^{H \times W \times C_h}$, the patch-embedding layer segments it into $N$ non-overlapping sub-images, denoted as $\boldsymbol{X} = [\boldsymbol{x}_1, \boldsymbol{x}_2, ..., \boldsymbol{x}_N]$ where each $\boldsymbol{x} \in \mathbb{R}^{K \times K \times C_h}$. Here, $K$ is the patch size, and $N = (H//K) \times (W//K)$ denotes the total number of patches. The DMD patterns used for computation are expected to be of identical spatial dimensions, $i.e.$, $\boldsymbol{\theta} \in \{0,1\}^{K \times K}$. To assimilate data across all spectral channels and ensure adaptive interaction of each convolutional kernel with various spectral channels, we introduce a learnable vector $\boldsymbol{v} \in \mathbb{R}^{C_h}$ for every convolutional kernel.

This convolutional approach, entailing single-channel convolutions followed by aggregating multi-channel outputs, bears resemblance to depthwise separable convolution, albeit with distinctions. In pursuit of enhanced efficiency, we opt to leverage the DMD to execute convolution across multiple spectral channels employing a singular-channel convolutional kernel in a single DMD$(\cdot, \cdot)$ operation.

Given the patch-embedding has $C$ convolutional kernels, we denote the sequence of these kernels as $\boldsymbol{\Theta} = [\boldsymbol{\theta}_1, \boldsymbol{\theta}_2, ..., \boldsymbol{\theta}_C]$. Similarly, the spectral weight vectors corresponding to these kernels are represented by $\boldsymbol{V} = [\boldsymbol{v}_1, \boldsymbol{v}_2, ..., \boldsymbol{v}_C]$. Hence, for an input image $\boldsymbol{X}$, the output $\boldsymbol{Y}$ from the patch-embedding layer can be expressed as:

$$\boldsymbol{y}_i = [\text{DMD}(\boldsymbol{x}_i, \boldsymbol{\theta}_1)\boldsymbol{v}_1^T, \text{DMD}(\boldsymbol{x}_i, \boldsymbol{\theta}_2)\boldsymbol{v}_2^T, ..., \text{DMD}(\boldsymbol{x}_i, \boldsymbol{\theta}_C)\boldsymbol{v}_C^T] \in \mathbb{R}^C, \quad i \in \{1, 2, ...N\},$$
$$\boldsymbol{Y} = [\boldsymbol{y}_1, \boldsymbol{y}_2, ..., \boldsymbol{y}_N]^T \in \mathbb{R}^{N \times C}. \tag{2}$$

In this computation, $\boldsymbol{Y}$ is obtained by $N \times C$ instances of the DMD$(\cdot, \cdot)$ operation. Given the constant efficiency inherent to the DMD$(\cdot, \cdot)$ operation, apparently, the number of DMD$(\cdot, \cdot)$ operations to compute $\boldsymbol{Y}$ solely influence the under-sampling rate. Thus, it is the optimization target.

## 3.2 LEARNABLE UNDER-SAMPLING MASK

To achieve under-sampling, we employ a binary decision mask with trainable parameters. In the Training Phase, these parameters directly determine the values of the mask, thereby affecting the loss function values of two metrics: the accuracy of downstream tasks and the under-sampling rate. Consequently, these parameters are updated during backpropagation, ultimately yielding a mask that meets the undersampling rate criteria and is optimally adapted to the downstream tasks.

Specifically, The binary decision mask $\boldsymbol{D} \in \{0,1\}^{N \times C}$ determines whether to retain or bypass each patch for a specific convolutional kernel. If the element at the $i$-th row and $j$-th column of the matrix $\boldsymbol{D}$ is 0, then the DMD will bypass $\boldsymbol{x}_i$ in optical modulation when using convolutional kernel $\boldsymbol{\theta}_j$. $\boldsymbol{D}$ sparsifies $\boldsymbol{Y}$, effectuating under-sampling in optical computations:

$$\boldsymbol{Y} \leftarrow \boldsymbol{Y} \odot \boldsymbol{D}. \tag{3}$$

The count of points passing through the mask corresponds to the number of DMD$(\cdot, \cdot)$ operations required to get $\boldsymbol{Y}$, equating to the summation of $\boldsymbol{D}$. We define the passing-through ratio as $d_{ops}$, where $d_{ops} = \text{sum}(\boldsymbol{D})/(N \times C)$.

The trainable parameters $\boldsymbol{z} \in \mathbb{R}^{N \times C \times 2}$ determines the value of $\boldsymbol{D}$. $\boldsymbol{z}$ undergoes a Linear layer followed by a Softmax operation to to assign probabilities to the elements of $\boldsymbol{D}$:

$$\boldsymbol{\pi} = \text{Softmax}(\text{Linear}(\boldsymbol{z})) \in \mathbb{R}^{N \times C \times 2}, \tag{4}$$

where $\boldsymbol{\pi}_{i,j,0}$ denotes the probability of bypassing the $i$-th patch for the $j$-th convolution kernel, and $\boldsymbol{\pi}_{i,j,1}$ represents the probability of retaining it. Thus, $\boldsymbol{D}$ can be sampled on the probabilities provided by $\boldsymbol{\pi}$. We deem the Linear layer here crucial since removing it led to a performance decline.

However, sampling from $\boldsymbol{\pi}$ is non-differentiable, which hinders the end-to-end training. To address this problem, we employed the Gumbel-Softmax technique (Jang et al., 2017) to sample from $\boldsymbol{\pi}$:

$$\boldsymbol{D} = \text{Gumbel-Softmax}(\boldsymbol{\pi})_{*,*,1} \in \{0,1\}^{N \times C}, \tag{5}$$

where the Gumbel-Softmax operation yields a two-element one-hot vector for each position in $\boldsymbol{D}$; one-hot vector $[0,1]$ indicates value 1 and $[1,0]$ value 0. The probabilistic expectation of $[0,1]$ equals $\boldsymbol{\pi}$. Through the Gumbel-Softmax technique, sampling from $\boldsymbol{\pi}$ transforms into a mathematical operation involving $\boldsymbol{\pi}$ and values sampled from a uniform distribution. This alteration makes the sampling process differentiable to $\boldsymbol{\pi}$, thus enabling end-to-end training.

During the network's finetuning process, $z$ is trained concurrently, and the under-sampling rate is integrated into the training objective using the Mean Squared Error (MSE) loss. Denoting the target mask ratio as $d_{tar}$, we have:

$$\mathcal{L}_{\text{ratio}} = \frac{1}{B} \sum_{b=1}^{B} (d_{tar} - d_{ops})^2, \tag{6}$$

where $B$ denotes the batch size. The total training loss merges the mask loss and the classification loss as follows:

$$\mathcal{L} = \mathcal{L}_{cls} + \lambda_{ratio}\mathcal{L}_{ratio}, \tag{7}$$

where $\mathcal{L}_{cls}$ denotes the classification loss. We set $\lambda_{ratio} = 5$ in our experiments.

### 3.3 KERNEL-LEVEL WEIGHT BINARIZATION

The DMD exclusively performs spatial binary modulation, necessitating a single-channel binary pattern $\theta$ in the patch-embedding layer. To ensure DMD compatibility, the patch-embedding layer's weights warrant binarization. Note that the binarization here is intended to adapt to DMD optical computation, not for network lightweighting.

Binarization in deep learning, such as in Binary Neural Networks (BNN)(Qin et al., 2020), simplifies models for reduced inference requirements. Typically, this involves binarizing weights and sometimes dataflow. Notable efforts in Transformer binarization include BiT(Liu et al., 2022), BinaryViT (Xiao et al., 2023) and BiSCI (Cai et al., 2023).

Due to the substantial performance degradation caused by full-layer binarization, a more refined strategy is required. As our binarization objective is to conform to the DMD operational constraints, it's adequate to ensure the pattern used in a single DMD($\cdot, \cdot$) operation is binarized, i.e., $\theta \in \{0, s\}^{K \times K}$ ($s \in \mathbb{R}$ elaborated upon in the following), instead of binarizing entirely. Specifically, the parameter matrix that determines the values of the convolutional kernels in the patch-embedding layer is represented as $W \in \mathbb{R}^{K \times K \times C}$, with its binarized outcome being $W^b \in \{0, 1\}^{K \times K \times C}$. The binarization process is realized through the step($\cdot$) function:

$$w^b = \text{step}(w) = \begin{cases} 1, & w \geq 0, \\ 0, & w < 0, \end{cases} \tag{8}$$

where $w \in W$ and $w^b \in W^b$. Let $w_i \in \mathbb{R}^{K \times K}$ be the parameter matrix determining each convolutional kernel in $W$, for $i \in \{1, 2, ..., C\}$, and let $w_i^b \in \{0, 1\}^{K \times K}$ be its corresponding binary matrix, we compute a weight coefficient $s \in \mathbb{R}$ for each convolutional kernel:

$$s_i = \frac{1}{K^2} \sum_{j=1}^{K} \sum_{k=1}^{K} \max(0, (w_i)_{jk}), \quad i \in \{1, 2, ..., C\}. \tag{9}$$

Thus, a single binarized convolutional kernel $\theta$ can be computed by:

$$\theta_i = s_i w_i^b, i \in \{1, 2, ..., C\}. \tag{10}$$

Despite optimizations, binarization introduces certain training challenges. Specifically, LUM-ViT, being a binarized-full precision hybrid model, necessitates distinct training configurations for each precision type. Furthermore, as the binary layer forms the network's foundation, alterations herein significantly affect subsequent layers, especially the mask training. Attempts to concurrently train the binary layer and the mask have thus been unsuccessful. To address this, we employed a three-stage fine-tuning strategy. Specifically, we utilize the MAE (He et al., 2021) base model as the pre-trained model and train LUM-ViT for the following three stages:

**Stage 1:** Initially, we trained the mask without binarization, using $w$ in place of $\theta$, employing standard AdamW (Loshchilov & Hutter, 2019) optimizer settings and regular data augmentations. The mask weights were then locked to ensure stability in later stages.

**Stage 2:** Binarization was introduced and trained, deriving $\theta$ from $w$. We utilized a more sensitive AdamW setting, minimizing data augmentations for this phase.

**Stage 3:** With the mask and binarization layer frozen, we fine-tuned subsequent layers, reverting to conventional AdamW settings and standard data augmentation strategies.

This three-stage training scheme yields favorable performance. For more detailed training settings, please refer to Section E of the Appendix.

# 4 EXPERIMENTS

In this section, we demonstrate LUM-ViT's performance. In section 4.1, we compare LUM-ViT against baseline methods and perform ablation studies on its design choices on the ImageNet-1k dataset. In section 4.2, real-world experiments on the DMD signal acquisition system are conducted to demonstrate LUM-ViT's real-world feasibility. In section 4.3, we apply LUM-ViT to three widely used hyperspectral image classification datasets, validating its utility for hyperspectral data.

## 4.1 THE TRAINING PHASE EXPERIMENTS

**(a)** Top-1 Acc before and after bianrization.

| $d_{tar}(\%)$ | LUM-ViT Top-1 Acc (%) | |
|---|---|---|
| | before bi | after bi |
| 20 | 82.7 | 82.5 |
| 10 | 82.3 | 81.9 |
| 5 | 80.0 | 79.8 |
| 2 | 78.9 | 78.3 |

**(b)** Mean keep probabilities of each patch position in LUM-ViT-2%.

**(c)** Comparison with basic methods.

Figure 4: **The main results in the Training Phase of LUM-ViT,** providing the Top-1 Acc results on the ImageNet-1k classification task. Comparisons with the basic methods reveals the reliability of the learnable under-sampling mask strategy, especially at extremely low under-sampling rates (2%-5%). The dark red line marks the baseline upper bound.

**Dataset.** Our experiments are conducted on ImageNet-1k (Deng et al., 2009), reporting the Top-1 classification accuracy. We refrained from utilizing hyperspectral datasets for their lack of comparability in comprehensiveness and richness to ImageNet. Additionally, the scarcity of real-world display devices for rendering hyperspectral images presents a major challenge for conducting practical experiments with such datasets.

**Pre-trained Model.** Our three-stage training begins with MAE-base as the pre-trained model. Its performance, with an accuracy of 83.7%, serves as our baseline upper bound. Following the MAE-base configuration, we set the patch size $K = 16$ and the number of kernels $C = 768$ in LUM-ViT.

**Performance and Training Details.** We trained LUM-ViT models separately for under-sampling rates of 20%, 10%, 5%, and 2%. Fig. 4 (a) shows their performance after Stage 1 and Stage 3, showcasing the learnable mask's impact and influence of binarization. Our training, detailed in Section 3.3, consists of three stages with 50, 20, and 80 epochs, respectively, including a warm-up for 1/10 of the total epochs per stage. In stages 1 and 3, we used AdamW optimizer with momenta 0.9 and 0.999 while adjusting to 0.6 and 0.9999 in stage 2. Common augmentations were used in stages 1 and 3, omitting cutmix (Zhong et al., 2020) and mixup (Zhang et al., 2018) in stage 2.

**Methods for Comparison.** Finding comparable analogous DL methods for benchmarking LUM-ViT is challenging as it's a pioneer in its field. CS methods, unsuitable for benchmarks, struggle with very low under-sampling rates. Given these circumstances, as a compromise, we introduce a method called DU-ViT that attempts **d**irectly **u**ndersampling by decreasing the number of convolutional kernels, serving as a comparative baseline. To further validate the effectiveness of the learnable mask strategy, we also experimented with random masking (Random-mask-ViT) and magnitude-based masking (Mag-mask-ViT), both of which showed a significant gap compared to the learnable mask strategy. Detailed descriptions of the comparative methods can be found in Section B of the Appendix.

**Performance Analysis.** Fig. 4 (c) depicts LUM-ViT surpassing DU-ViT at all under-sampling rates, notably maintaining accuracy loss below 5.5% at extremely low rates (2%-5%). This reflects that the original image holds more information than needed for classification, with the learnable mask adept at isolating vital information for the task. Fig. 4 (b) illustrates the average pass-through probability across all patch positions for LUM-ViT-2% (LUM-ViT with 2% mask). No global center-edge pass-through rate disparity was observed, indicating that a fixed mask solely focusing on the image center may be inadequate for this dataset. For further analysis of the mask, please refer to Section C of the Appendix.

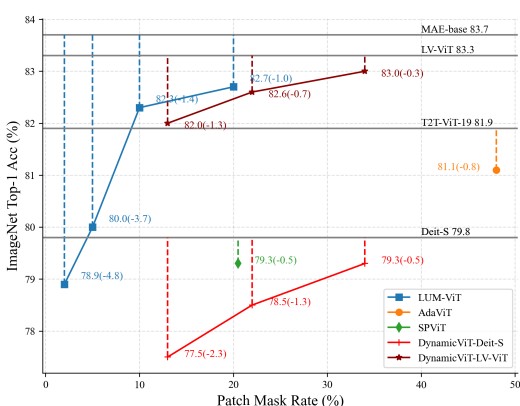

Figure 5: **Comparison of mask patch ratios.** This is not a performance comparison for network lightweighting, but to prove the effectiveness of the learnable mask of LUM-ViT.

**Mask Analysis.** By pruning patch information to 2% through a mask before the backbone, LUM-ViT accomplishes classification tasks, aligning with the notion that the inputs excess information for classification. Fig. 5 contrasts LUM-ViT with other mask patch methods on ViT, indirectly substantiating the effectiveness of the learnable mask. Notably, this is not a performance comparison. Indeed, LUM-ViT and these methods are not directly comparable in performance, given that LUM-ViT does not aim to lightweight the backbone.

**Ablation Study.** Four scenarios were explored in ablation studies on LUM-ViT-10% (LUM-ViT with 10% mask) to validate our methodology. Each setup underwent thorough training, reporting Top-1 accuracy pre and post-binarization (post Stage-1 and Stage-3 training). Here, **full-layer bi** indicates binarizing at layer-level rather than kernel-level. **W/o learnable token** denotes zero assignment to masked points instead of learnable token substitution. **L1-norm** involves using L1-norm for loss calculation during mask training, replacing MSE. **No Linear** signifies excluding the Linear layer following the masking parameters.

Table 1: **Results of the ablation study.**

| Config | Top-1 Acc (%) | |
| --- | --- | --- |
| | before bi | after bi |
| **LUM-ViT** | **82.3** | **81.9** |
| full-layer bi | 82.3 | 81.3(-0.6) |
| w/o learnable token | 82.1(-0.2) | 81.7(-0.2) |
| L1-norm | 80.6(-1.7) | 80.0(-1.9) |
| no Linear | 79.7(-2.6) | 79.4(-2.5) |

## 4.2 THE REAL-WORLD APPLICATION PHASE EXPERIMENTS

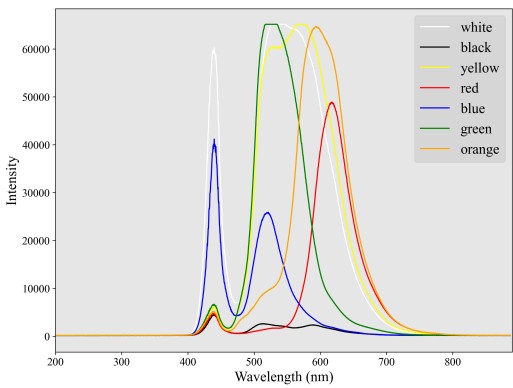

(a)Spectrum of each color in the e-ink display.

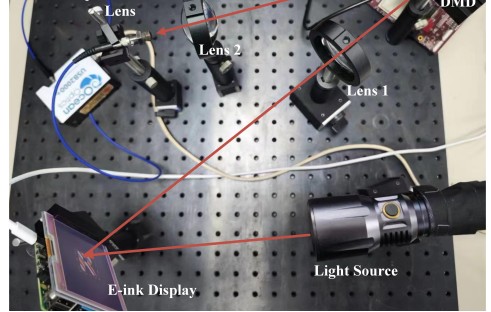

(b)Real-world DMD signal acquisition system.

Figure 6: **Real-world DMD signal acquisition system and spectrum of each color in the e-ink display.** We established a setup in line with the standard DMD signal acquisition system. To ensure spectral integrity, images are displayed on an e-ink screen with a stable LED light source employed for illumination.

**Hardware Setup.** Our configuration comprises a **DMD** from Texas Instruments (resolution: 1024*768 pixels), an Ocean Optic **Spectrometer** (spectral range: 177.1-886.6 nm across 2024

channels), and a stable **LED** for illumination. For **object display**, given the need for natural spectra display and adequate light projection onto the DMD, we forwent a conventional monitor in favor of a 7-color e-ink display, which reflects natural spectra under robust light source illumination.

**Model Adjustment.** The LUM-ViT model trained in the Training Phase cannot directly deploy on real-world hardware. Images from ImageNet comprise merely three color channels, *i.e.*, $C_h = 3$, whereas the high-spectral data encompasses 2048 color channels, *i.e.*, $C_h = 2048$. To address this discrepancy, we reconfigured the spectral channel weighting parameter $V$ (based on the spectrum information illustrated in Fig. 6 (a)) to establish a new color correspondence and conducted fine-tuning on a small sample set.

**Real-world Experiment Results.** We conducted a real-world application experiment on LUM-ViT-2%. Our basic hardware setup, including a 7-color e-ink display, led to significant accuracy loss. We argue that this loss, being device-induced, should be isolated from our method's performance evaluation. Thus, we color-reduced ImageNet images to 7 colors in software simulation, reporting the accuracy loss across different models. Certain samples, initially classified by MAE, failed post-color-reduction and were marked as device-induced anomalous samples.

In the real-world application experiment, we processed 371 random images from the ImageNet-1k test set using LUM-ViT, with DMD implementation in computation. A total of 237 images were correctly classified, yielding a 64.4% accuracy. Among the misclassified images, 51 were device-induced anomalous samples. Excluding these, the accuracy improves to 74.7%. Therefore, the LUM-ViT model with 2% masking demonstrates practical feasibility in a real-world scenario with DMD involvement, keeping the accuracy loss within 4% for RGB images and within 5% for 7-color images. This showcases its real-world feasibility.

Table 2: **Results of the real-world experiments on LUM-ViT-2%.** Accuracy degradation induced by 7-color conversion is reported.

| Model | Top-1 Acc (%) | |
|---|---|---|
| | RGB | 7-color |
| MAE-base | 83.7 | 75.7 |
| LUM-ViT-2% | 78.3 | 69.1 |
| LUM-ViT-2%(real-world) | 74.7 | **64.4** |

### 4.3 THE HYPERSPECTRAL IMAGE CLASSIFICATION EXPERIMENTS

To ascertain whether LUM-ViT can handle real-world hyperspectral data with richer spectral information, we evaluated its performance on three remote sensing datasets widely used in hyperspectral image classification tasks: Indian Pines (Baumgardner et al., 2015), Pavia University, and Salinas. As indicated in Table A, LUM-ViT achieved commendable results (Performance is measured by the proportion of correctly classified samples (Overall Accuracy)). Notably, this experiment aimed to verify LUM-ViT's efficacy in handling datasets with abundant spectral channel information, complementing the ImageNet-1k dataset classification task. For detailed experimental design, please refer to Section A of the Appendix.

Table 3: **Results on HSI classification.**

| Dataset | Overall Accuracy (%) | |
|---|---|---|
| | before mask | after mask |
| Indian Pines | 87.4 | 86.2 |
| Pavia University | 89.5 | 88.6 |
| Salinas | 99.4 | 99.1 |

## 5 CONCLUSION

In this work, we innovatively applied deep learning to under-sample hyperspectral data acquisition, achieving significant data reduction from signal collection to processing. Utilizing ViT as the backbone network and a DMD signal acquisition system for patch-embedding, we enabled pre-acquisition optical modulation. With a learnable mask, we identified and retained embedded points crucial for downstream tasks while bypassing less significant embedded points, achieving under-sampling. On the ImageNet-1k classification task, we maintained accuracy loss within 1.8% at 10% under-sampling and within 5.5% at an extreme 2% under-sampling. Real-world experiments showed that the accuracy loss of LUM-ViT did not exceed 4% compared to the software environment, demonstrating its practical feasibility. Future research could explore dynamic mask strategies and extend this framework to other tasks.

ACKNOWLEDGMENTS

This work was supported by National Natural Science Foundation of China under Grant 62173298. *(Corresponding author: Dong Ni.)*

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

# APPENDIX

## A  THE HYPERSPECTRAL IMAGE CLASSIFICATION EXPERIMENTS

### A.1  DATASETS

Due to the absence of a comprehensive hyperspectral dataset for object classification as extensive as ImageNet in the open-source domain, we opted for the next best alternative: three remote sensing datasets commonly used in hyperspectral image classification tasks—Indian Pines, Pavia University, and Salinas. Below is a detailed introduction to these three datasets.

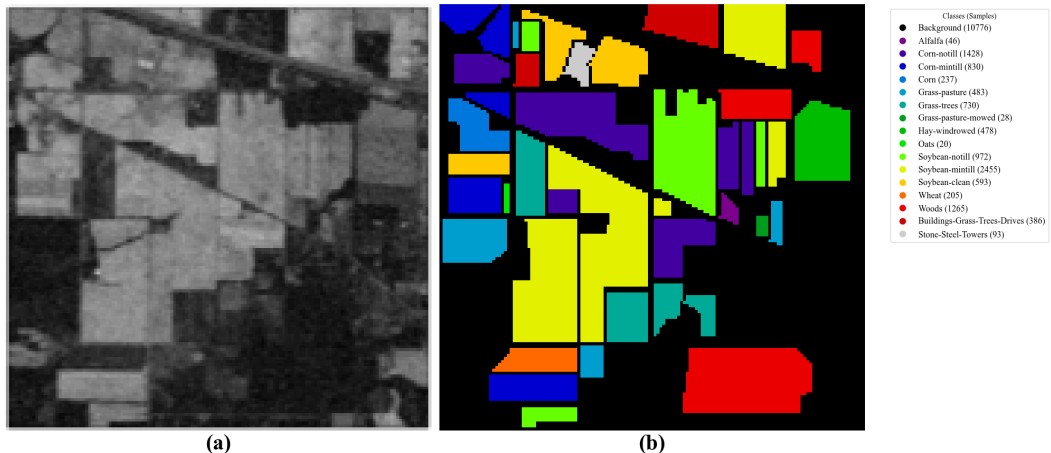

Figure 7: **Indian Pines dataset:** (a) Grayscale schematic diagram, (b) Ground truth labels (the pixel count for each sample type is displayed on the legend).

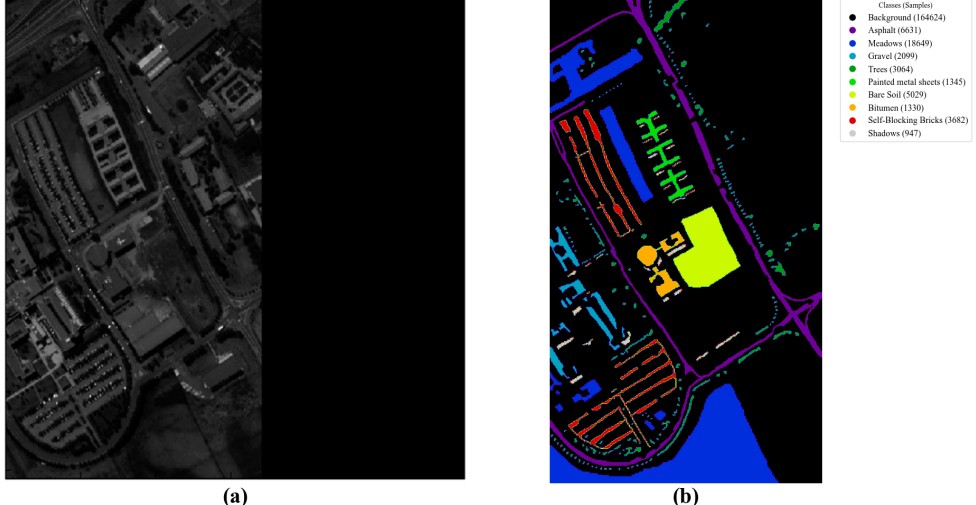

Figure 8: **Pavia University dataset:** (a) Grayscale schematic diagram, (b) Ground truth labels (the pixel count for each sample type is displayed on the legend).

**Indian Pines (Fig. 7):** The AVIRIS sensor captured the Indian Pines scene over a test site in Northwestern Indiana, which includes a 145x145 pixel array with 224 spectral reflectance bands spanning the 0.4–2.5 micrometer wavelength range. This dataset is a subset of a larger scene, predominantly composed of two-thirds agricultural land and one-third forest or other perennial natural vegetation. The area features significant infrastructure, including major highways, a railway, low-density housing, other built structures, and minor roads. Captured in June, the image shows early-stage crops like

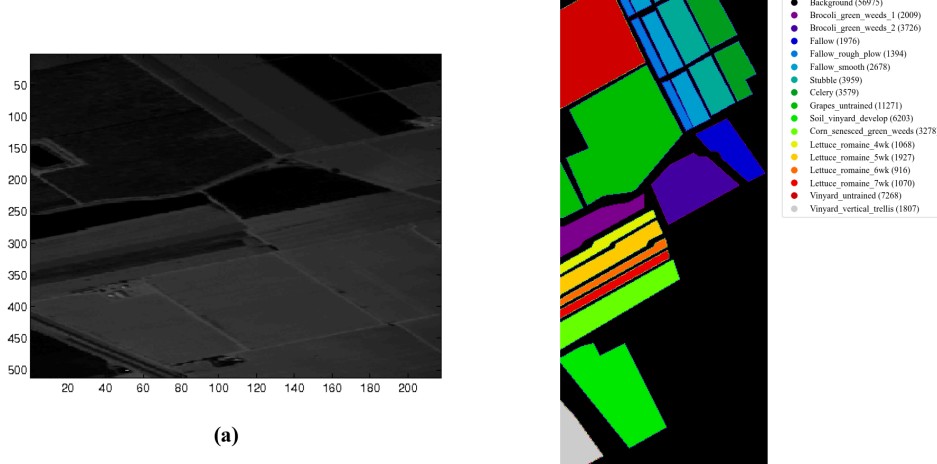

Figure 9: **Salinas dataset:** (a) Grayscale schematic diagram, (b) Ground truth labels (the pixel count for each sample type is displayed on the legend).

corn and soybeans with less than 5% coverage. The provided ground truth is divided into sixteen classes, which are not entirely distinct. To avoid water absorption features, the number of spectral bands has been reduced to 200 by excluding bands [104-108], [150-163], and 220.

**Pavia University (Fig. 8):** This scene was captured by the ROSIS sensor during a flight campaign over Pavia in Northern Italy. It comprises 103 spectral bands. The Pavia University image is 610 by 610 pixels, though some samples in both images are void of information and must be discarded prior to analysis. The geometric resolution of the image is 1.3 meters. The image ground truths distinguish nine different classes. The figures show the discarded samples as wide black strips.

**Salinas (Fig. 9):** The Salinas scene was collected using the 224-band AVIRIS sensor over Salinas Valley, California, noted for its high spatial resolution with 3.7-meter pixels. The captured area includes 512 lines by 217 samples. Similar to the Indian Pines scene, 20 water absorption bands were omitted, specifically bands: [108-112], [154-167], and 224. This imagery was only available in the form of at-sensor radiance data. The scene features a variety of elements including vegetables, bare soils, and vineyard fields. The Salinas ground truth encompasses 16 classes.

## A.2 TASK DESCRIPTION AND EXPERIMENTAL SETUP

We conducted pixel classification tasks on these three datasets, classifying each pixel based on its own spectral data as well as that of its surrounding pixels. Although there is no universally accepted standard for dividing training and validation data, we followed the conventional approach of a large validation set and a small training set (6:4), ensuring no overlap between training and validation data. The dataset partitioned for this task comprises samples, each a 9x9 sub-image consisting of the target pixel for classification and its surrounding pixels.

To tailor to this task, we adjusted the configuration of LUM-ViT: the number of input channels was matched to the spectral channels of the hyperspectral datasets, the patch size was set to 9, and images input into the network were first enlarged to 27x27, resulting in 9 patches. Additionally, we reduced the embedding dimension to 192, while the rest of the settings remained largely consistent with those used for ImageNet.

We conducted training for 100 epochs from scratch on all three datasets, including a 5-epoch warmup, with a batch size of 64 and a peak learning rate of 0.01, utilizing cosine annealing. We observed some degree of overfitting and made fine adjustments accordingly. A large weight decay (0.001) setting could mitigate this issue.

## B  BASE METHODS FOR COMPARISON

In the experimental section, we compared LUM-ViT with DU-ViT, Random-mask-ViT, Mag-mask-ViT and CS-method on the ImageNet-1k classification task to demonstrate the superior performance and design of LUM-ViT. This section will introduce the experimental setup for these four methods.

### B.1  DU-ViT

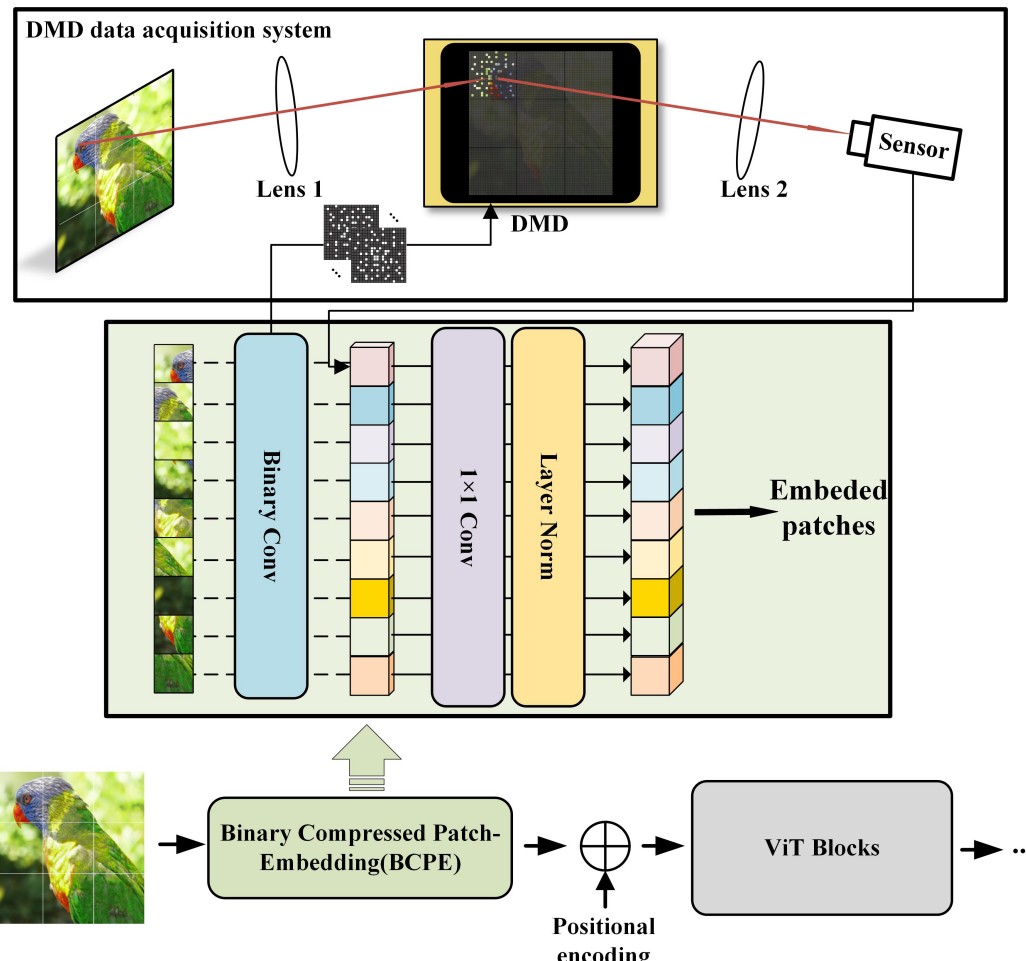

Figure 10: **DU-ViT.** We integrated the design of under-sampling and binary convolution within the patch-embedding layer, termed the Binary Compressed Patch-Embedding (BCPE) layer. The dashed lines in the diagram represent data flows exclusive to the training phase.

In early experiments, we considered a straightforward method to reduce the number of convolutional kernels in the patch-embedding layer to achieve under-sampling. We experimented with this method but observed poor performance. We speculate that the reduction in convolutional kernels led to a significant decrease in the embedded dimension of the data input to the backbone, and this reduction in data dimensionality severely limited the model's expressive capability. Building on this, we added $1 \times 1$ convolutional layer after the reduced-kernel convolution to restore the embedded dimension to its pre-under-sampling size. We refer to this network as DU-ViT, that is, Directly Under-sampling ViT. This network achieved reasonably good results in our experiments and can serve as a baseline model for under-sampling.

The structure of DU-ViT is shown in Fig. 10. To accommodate DMD operations, this method also employs binary convolutional layers. In our experiments, we primarily observed that adding a

Layer Normalization after the $1 \times 1$ convolution further enhances performance, hence this setup was retained. During the training of DU-ViT, we maintained most of the training settings consistent with LUM-ViT to facilitate comparison, including the three-stage training strategy.

We noted that the performance difference between DU-ViT and LUM-ViT remains relatively stable when the under-sampling rate is above 10%. However, under extreme under-sampling conditions, such as 2%, there is a significant gap in performance between DU-ViT and LUM-ViT.

## B.2 RANDOM-MASK-VIT AND MAG-MASK-VIT

We believe that LUM-ViT's ability to maintain performance at low under-sampling rates is not solely due to the adoption of a masking approach for under-sampling, but also attributed to the success of the learnable mask strategy. To confirm the superiority of the learnable mask approach, we conducted experiments with other masking strategies, summarized into the following two schemes:

**Random-mask-ViT:** This approach utilizes a random mask strategy. Specifically, we randomly generated a mask and applied it to the output of the patch-embedding layer in the pre-trained model, followed by finetuning with the mask fixed in place.

**Mag-mask-ViT:** Drawing on common strategies from neural network pruning, we implemented a mask based on data magnitude. Specifically, we calculated the average magnitude at each position of the patch-embedding layer's output from the pre-trained model using the training dataset. We then selected the positions with the highest magnitudes to retain according to the required under-sampling ratio, masking the rest. With the mask fixed, we proceeded to finetune the model.

Under the condition that most training settings remained unchanged, we trained and tested the Random-mask-ViT and Mag-mask-ViT models. We observed that the performance of these two schemes was inferior to DU-ViT, and even less so to LUM-ViT. Similar to DU-ViT, we also noted a more rapid decline in performance at extreme under-sampling rates (2%). Based on the comparison of LUM-ViT with these two models, we conclude that the learnable mask strategy with trainable parameters is superior, adapting well to the target datasets and downstream tasks.

## B.3 CS-METHOD

Compressed Sensing (CS) is an innovative signal acquisition and reconstruction theory that allows for the capture of signals at a rate significantly below the Nyquist sampling criterion and accurately reconstructs sparse signals or signals that can be sparsely represented from these few samples.

The principle of CS indicates that for a sparse signal, or a signal that can be sparsely represented, it is possible to capture most of the signal information through low-dimensional measurements and to accurately reconstruct the original signal from these measurements. This process can be described by the following mathematical model:

$$\mathbf{y} = \Phi \mathbf{x}, \tag{11}$$

where $\mathbf{x} \in \mathbb{R}^n$ is the original signal, $\Phi \in \mathbb{R}^{m \times n}$ is the measurement matrix, and $\mathbf{y} \in \mathbb{R}^m$ are the measurements with $m \ll n$, meaning that the dimension of measurements is much less than the dimension of the signal.

The sparsity of a signal refers to the number of non-zero elements in a transform domain (like Fourier, wavelet, *etc.*) being few. If the signal itself is not sparse, it can be transformed using a transformation matrix $\Psi$:

$$\mathbf{s} = \Psi \mathbf{x}, \tag{12}$$

where $\mathbf{s}$ is the sparse representation in the transform domain.

For signal reconstruction, one of the common algorithms is Orthogonal Matching Pursuit (OMP). OMP is an iterative greedy algorithm that seeks to find the set of sparse coefficients that best matches the measurement vector $\mathbf{y}$. The steps of the OMP algorithm are as follows:

1. Initialize the residual $\mathbf{r}_0 = \mathbf{y}$, support set $S_0 = \emptyset$, and iteration count $k = 0$.
2. Find the column $\phi_j$ that correlates most significantly with the residual $\mathbf{r}_k$ and update the support set $S_{k+1} = S_k \cup \{j\}$.

3. Update the signal estimate by solving the least squares problem: $\mathbf{x}_{k+1} = \arg\min_{\mathbf{x}} \|\mathbf{y} - \Phi_{S_{k+1}}\mathbf{x}\|_2$.

4. Update the residual to $\mathbf{r}_{k+1} = \mathbf{y} - \Phi_{S_{k+1}}\mathbf{x}_{k+1}$.

5. If a stopping criterion is met (such as the residual is small enough or a predetermined number of iterations is reached), stop the iteration.

6. Otherwise, increment $k$ and go back to step 2.

Based on the aforementioned methods, we conducted tests on the ImageNet-1k dataset. Specifically, we undersampled the original images at different under-sampling rates using a Gaussian random mask (this process is adapted for DMD computation), then reconstructed the images using the OMP algorithm, and finally completed the classification task through ViT.

Because the CS-method yields poor reconstruction of the original image at low under-sampling rates ($< 20\%$), the subsequent ViT model struggles with the classification task, leading to very poor final results. The contrast between these results and those of LUM-ViT aligns with our logic: the CS-method cannot effectively utilize the prior information of the object being detected, making it unsuitable for low under-sampling rates, whereas deep learning methods can achieve better results by effectively using prior information, maintaining performance even at extremely low under-sampling rates.

## C  VISULIZATION OF THE LEARNABLE MASK AND ANALYSE

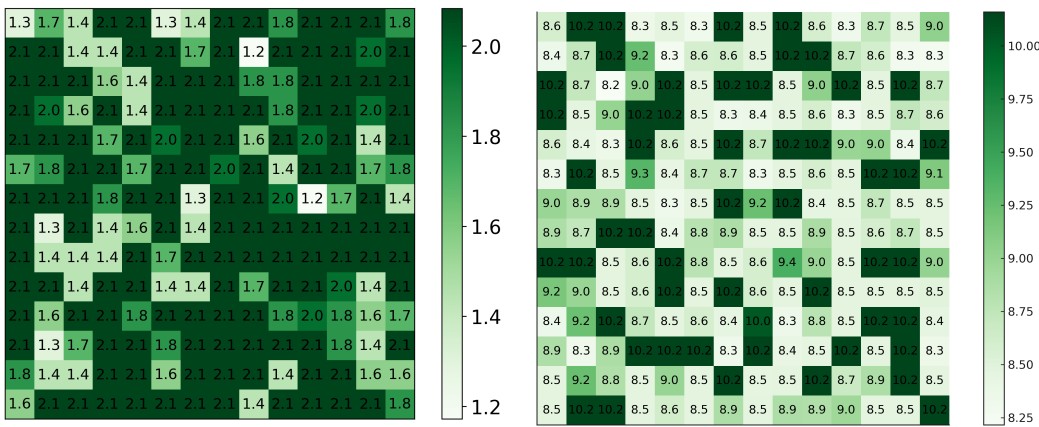

(a)Heatmap for LUM-ViT-2% patches.  (b)Heatmap for LUM-ViT-10% patches.

Figure 11: **The passing-throgh probability heatmap of patches at different positions in LUM-ViT.** In figure (b), we truncated the top 5% of the data to allow the image to reveal meaningful information and to prevent the distortion caused by extreme outliers.

In LUM-ViT, the mask is applied to each data point output by the patch-embedding layer, with each data point corresponding to the output of a specific convolutional kernel on a specific patch. The mask filters both kernels and patches: if all positions of patches affected by a kernel are masked, then the kernel is eliminated; similarly, if all kernel positions in a patch are masked, the patch is eliminated.

To explore the selective effect of the learnable mask on different kernel-patch data points, we visualized two models: LUM-ViT-10% and LUM-ViT-2%.

Fig. 11 presents the visualization results for different patch positions, offering an intuitive realization of the macroscopic emphasis of all kernels on different image locations. Overall, we did not observe a pronounced spatial preference, which might be due to the fact that all patches could contain important information across the entire dataset. Comparing the two models, the heatmap of LUM-ViT-10% appears more random, and we observed four extreme outliers with values exceeding 60%, which, if not truncated, would significantly impact the visualization effect.

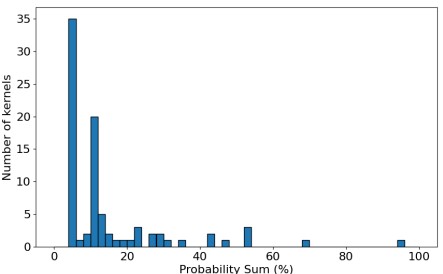 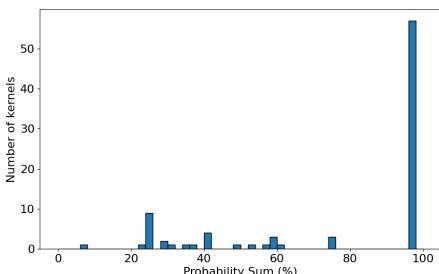

(a)Histogram for LUM-ViT-2% kernels.    (b)Histogram for LUM-ViT-10% kernels.

Figure 12: **Histogram of the number of patches retained by each convolutional kernel in LUM-ViT.** The horizontal axis represents the sum of probabilities for all patch positions in a single kernel to pass through the mask. Due to the selection and elimination of most convolutional kernels in low under-sampling scenarios, the number of data points at position 0 in the histogram is overwhelmingly large. To prevent this from impacting the visual effect, we have excluded the data points within the [0,5] range.

Fig. 12 visualizes the selection probability of different kernels. We calculated the sum of probabilities for all patch positions passing through the mask for each kernel and tallied the number of kernels for different total probability sums. Many kernels are directly discarded at low under-sampling rates, leading to an overwhelming number of points near 0. To avoid impacting the visualization, we removed data within the [0,5] interval. We observed that in LUM-ViT-10%, there is a tendency to select kernels without selecting patches, meaning the model tends to keep or discard all patches under one kernel. This phenomenon disappears in LUM-ViT-2%, where the model tends to allocate different patch positions to different kernels.

Synthesizing the visual results from Fig.11 and Fig.12, we infer that the model tends to select kernels under relatively low under-sampling pressure (*e.g.*, 10%), satisfying under-sampling requirements by retaining only the most useful kernels. When the under-sampling pressure is relatively high (*e.g.*, 2%), selecting kernels alone does not meet the under-sampling rate requirements, thus forcing the model to make selections among different patch positions under different kernels, leading to a collaborative division of labor across different kernels for different patch positions.

# D    DETAILS ON THE DMD SIGNAL ACQUISITION SYSTEM

## D.1    DETAILED OVERVIEW OF DMD

Digital Micromirror Devices (DMDs) are sophisticated optical semiconductor devices that form the core of Digital Light Processing (DLP) technology. A typical DMD chip hosts an array of up to millions of micromirrors, each with dimensions in the order of a few micrometers.

The core functionality of a DMD lies in its ability to modulate light spatially. Each micromirror can be individually tilted by electrostatic forces, typically between $+12$ to $-12$ degrees, corresponding to on and off states, respectively. This tilt controls whether light is directed towards the projection lens (on) or absorbed by a light trap (off), thereby controlling the brightness of each pixel in the projected image.

The speed at which these mirrors switch affects the image's refresh rate, with current DMDs capable of switching at speeds of thousands of times per second.

DMDs are integral to various applications due to their high spatial resolution and rapid response time. Common uses include high-definition projectors for cinema and business presentations, where their ability to precisely control light ensures high-quality image projection.

In the field of hyperspectral data processing, DMDs enable the selective reflection of different wavelengths, facilitating the capture of spectral information for each pixel, which is critical for applications in remote sensing and material analysis. Furthermore, the potential of DMDs in optical neural networks is being explored, where they can modulate light in complex patterns to mimic neural connections, potentially leading to advancements in photonic computing and artificial intelligence.

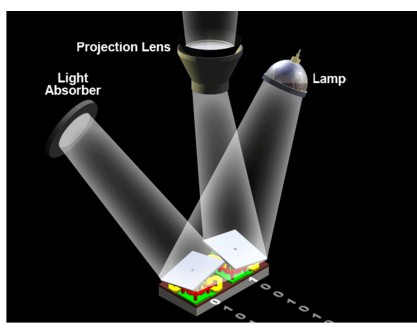
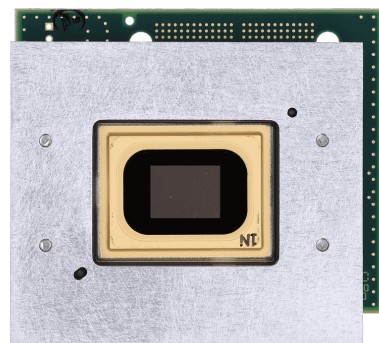



(a)The reflective function of a single
micromirror on a DMD chip.    (b)DMD evaluation board.

Figure 13: **DMD chip and its micromirror function.**



## D.2 DMD SIGNAL ACQUISITION SYSTEM

In LUM-ViT, the DMD serves as a key component for optical modulation before signal acquisition, termed the DMD signal acquisition system. Its schematic is shown in Fig. 1 in the main text, and its physical representation in Fig. 6, which are not displayed here again.

The DMD signal acquisition system comprises two sets of focusing lenses, a DMD chip, and a single-point light signal detector. During operation, light from the target object is imaged onto the DMD's micromirror surface by Lens Group 1. The DMD modulates the image projected onto it by setting its micromirrors to different reflection states. The modulated image is then reflected towards the optical detector, where Lens Group 2 focuses the light onto the detector, completing signal acquisition. It's worth noting that while the spectrometer here can capture comprehensive spectral information at an extremely high spectral resolution, it only measures the light intensity at a single point at a time, rather than capturing a multi-pixel image in one shot like conventional photographic equipment.

During its operation, the DMD can control the state of its micromirror array at each timestep, modulating the optical signal in various ways. Since the DMD performs spatial optical modulation—simultaneously modulating all spectral channels within its working range with the same mask—it is well-suited for modulating hyperspectral signals.

The DMD is the critical control component of the DMD signal acquisition system, with signal modulation achieved through different masks. In single-pixel cameras based on CS theory, the DMD typically uses random masks following a Gaussian distribution. In LUM-ViT, neural network weights obtained through training are used as masks for the DMD, integrating the DMD signal acquisition system into the neural network computation.

## E  TRAINING SETTINGS OF THE THREE STAGE OF THE TRAINING PHASE

During the Training Phase, we divided the training process into three stages, using distinct training parameters to train the mask, the kernel-level binary layer, and overall finetuning, respectively. In initial experiments, we attempted to combine these stages, but due to the instability introduced by the learnable mask and binarization, training them together led to difficulty in convergence or suboptimal performance. Therefore, we consider this three-stage training approach necessary.

Regarding the first stage of training, conventional training parameters suffice for mask training. The primary consideration is the initial value of the mask's trainable parameters. We experimented with an all-ones mask (i.e., a non-selective mask) as the initial value, but results were inferior to those with a randomly initialized mask. During training, the undersampling rate quickly drops to the target value within a few epochs and then fluctuates slightly throughout the remaining training process.

The second stage is the most unstable part of the entire training process. Due to the significant difference between binary and full precision, and because LUM-ViT requires binarization of the entire first layer (which affects all subsequent layers' computations), many settings from full-precision



19



network training are no longer applicable. Specifically, we froze the mask weights from the first stage to prevent changes in the mask from affecting the training of the binary layer. We abandoned most data augmentation strategies and adopted a more sensitive and stable learning rate momentum parameter setting, commonly used in training binary networks.

The third stage is the final finetuning. Since the second stage's training settings are primarily tailored to the binary layer and do not suit the full-precision layers, we need to finetune these layers last with parameters suitable for full-precision training. Specifically, in this stage, we froze the weights of the binary layer and the mask, finetuning only the remaining network layers with conventional parameter settings.

Table 4: Training settings.

| config | value | | |
|---|---|---|---|
| | **Stage 1** | **Stage 2** | **Stage 3** |
| optimizer | AdamW | AdamW | AdamW |
| optimizer momentum$(\beta_1, \beta_2)$ | 0.9, 0.999 | 0.6, 0.9999 | 0.9, 0.999 |
| base learning rate | 5e-4 | 1e-4 | 1e-4 |
| weight decay | 0.03 | 5e-5 | 0.03 |
| batch size | 4096 | 4096 | 4096 |
| learning rate schedule | cosine decay | cosine decay | cosine decay |
| warmup epochs | 5 | 1 | 5 |
| frozen layers | none | mask | bi-conv-kernels, mask |
| label smoothing | 0.1 | 0.1 | 0.1 |
| random erase | 0.25 | 0 | 0.25 |
| mixup | 0.8 | 0 | 0.8 |
| cutmix | 1.0 | 0 | 1.0 |
| drop path | 0.1 | 0.1 | 0.1 |

