# OpenReview forum: "LUM-ViT: Learnable Under-sampling Mask Vision Transformer for Bandwidth Limited Optical Signal Acquisition"
_ICLR.cc/2024/Conference — ICLR 2024 poster_

### Official Review · Reviewer_wWh6 · 2023-10-30

**Soundness:** 3 good
**Presentation:** 3 good
**Contribution:** 3 good
**Rating:** 6
**Confidence:** 5

**Summary:**

The proposed method of this paper is LUM-ViT, a learnable under-sampling mask vision transformer for bandwidth-limited optical signal acquisition. It is a novel approach that utilizes deep learning and prior information to reduce acquisition volume and optimize for optical calculations. The methodology unfolds in two primary stages: training from pre-trained models in a solely electronic domain using existing datasets, followed by inference to evaluate model performance, and assessing the real-world performance of LUM-ViT with a DMD signal acquisition system. During acquisition, target information undergoes a single instance of DMD optical modulation before capture, and then is funneled into the electronic system for further processing.

**Strengths:**

(i) The studied problem about under-sampling hyperspectral data acquisition, achieving data reduction from signal collection while preserving model performance. This accelerates the HSI processing in real applications such as remote sensing, object tracking, medical imaging, etc.

(ii) The idea of using a learnable mask refined during training to selectively retain essential points for downstream tasks from the patch embedding outputs, and thereby achieving under-sampling (reducing the required sampling instances) is interesting.

(iii) The performance is good. On the ImageNet-1k classification task, the proposed LUM-ViT maintains accuracy loss within 1.8% at 10% under-sampling and within 5.5% at an extreme 2% under-sampling.

(iv) This work not only conducts experiments on the synthetic data but also sets up hardware (as shown in Figure 6) to evaluate the effectiveness of the proposed method. It is a good and non-trivial exploration. The accuracy loss of LUM-ViT does not exceed 4% compared to the software environment, demonstrating its practical feasibility.

**Weaknesses:**

(i) The detailed formulations for DMD are missing, which is confusing. More explanations are required.

(ii) Code and pre-trained weights are not submitted. The reproducibility of this work cannot be checked.

(iii) The multi-stage training pipeline is tedious, which makes the whole technical route unreliable. What's worse, the finetuning details about stage 3 are missing.  Other researchers cannot re-implement this complex approach.

(iv) The writing should be further improved, especially the mathematical notations in Section 3.3. The formula for binary compression is not formal.

(v) The experiments are not sufficient and many critical comparisons are missing. For example, the backbone is ViT variants. However, the ViT is very computationally expensive because its computational complexity is quadratic to the input spatial size. This also embeds the real-time applications of HSI processing. In contrast, MST [1] or MST++ [2] are specially designed for HSI processing. They treat spectral feature maps as a token to capture the interdependencies between spectra with different wavelengths. Most importantly, they are very efficient with linear computational complexity regarding spatial resolution. Thus, it is better to add a comparison with the spectral Transformer in Figure 5.

(vi) The binarization mechanism is out of fashion. BiSCI [3] provides a specially designed binarized convolution unit BiSR-conv block to process HSI data cubes. It is also better to add a comparison with this new technique.

[1] Mask-guided Spectral-wise Transformer for Efficient Hyperspectral Image Reconstruction. In CVPR 2022.

[2] MST++: Multi-stage Spectral-wise Transformer for Efficient Spectral Reconstruction. In CVPRW 2022, NTIRE 2022 Winner in Spectral Recovery.

[3] Binarized Spectral Compressive Imaging. In NeurIPS 2023.

**Questions:**

The technical route and core idea in this paper is to accelerate the HSI processing, which is similar to Coded Aperture Spectral Snapshot Imaging (CASSI). Could you please analyze the differences, advantages, and disadvantages of these two systems?

---

> ### Author Response · Authors · 2023-11-20
> **Reply to reviewer wWh6**
>
> ## Reply to reviewer wWh6
>
> Dear Reviewer,
>
> We sincerely appreciate your professional and insightful review of our manuscript. Your detailed feedback has been immensely helpful and provided valuable guidance in improving our work. We are pleased with your interest in our learnable mask method and appreciate your acknowledgment of the LUM-ViT's performance. This acknowledgment serves as a significant encouragement to our team. Additionally, we value your appreciation and attention to our hardware work, which, though time-consuming and complex, is an indispensable part of our research.
>
> We apologize for the delay in our response. We have conducted additional experiments with two comparative models and on three real hyperspectral datasets. These experiments were time-consuming but have significantly contributed to making our experimental section more robust and convincing.
>
> Please note that all modifications in the manuscript are marked in red, except for the content in the Appendix, which is entirely new.
>
> We will now address each of the weaknesses and questions you raised in detail.
>
> ---
>
> > **W-1: The detailed formulations for DMD are missing, which is confusing. More explanations are required.**
>
> ---
> **REPLY**: Thank you for pointing out the need for a more detailed explanation of Digital Micromirror Devices (DMDs). DMDs are vital optical semiconductor devices that form the core of Digital Light Processing (DLP) technology. They consist of an array of up to millions of micromirrors, each with dimensions in the order of a few micrometers.
>
> The primary functionality of a DMD is its ability to spatially modulate light. Each micromirror can be individually tilted by electrostatic forces between \(+12\) to \(-12\) degrees, corresponding to on and off states, respectively. This tilt determines whether light is directed towards the projection lens (on state) or absorbed by a light trap (off state), thus controlling the brightness of each pixel in the projected image.
>
> While the DMD is a key component of the hardware system in our study, and the DMD signal acquisition system is central to optical computing and signal collection, this system is not the focus or the novelty of our work. Therefore, it was not elaborately detailed in the main manuscript. However, understanding the DMD system is fundamental to comprehending many designs of LUM-ViT. Acknowledging this, we have included a detailed explanation of the DMD system in Section D of the Appendix, considering the length constraints of the main text.
>
> ---
>
> > **W-2: Code and pre-trained weights are not submitted. The reproducibility of this work cannot be checked.**
>
> ---
> **REPLY**: We acknowledge your concern regarding the reproducibility of our work, particularly the absence of code and pre-trained weights. We are actively working on organizing and annotating the code for LUM-ViT's model construction and will make it available on the current platform as soon as possible. Furthermore, following the publication of our paper, we plan to open source all related codes and checkpoints on GitHub.
>
> This effort to provide detailed and well-documented code reflects our commitment to the scientific community's standards for reproducibility and transparency. We recognize the importance of your emphasis on these aspects and are dedicated to upholding these essential scientific principles.

---

> > ### Author Response · Authors · 2023-11-20
> >
> > ---
> >
> > > **W-3: The multi-stage training pipeline is tedious, which makes the whole technical route unreliable. What's worse, the finetuning details about stage 3 are missing. Other researchers cannot re-implement this complex approach.**
> >
> > ---
> > **REPLY**: We appreciate your concern regarding the complexity of the multi-stage training pipeline of LUM-ViT. LUM-ViT, as a binarized-full precision hybrid model, requires distinct training configurations for each precision type. The binary layer, forming the foundation of our network, significantly influences subsequent layers, particularly mask training. Our attempts to train the binary layer and mask concurrently have been unsuccessful, leading us to adopt a multi-stage training approach.
> >
> > We have experimented with an alternative strategy of binarizing first and then training the mask, but found that it did not yield results as effective as our current three-stage training process. Therefore, we consider the three-stage training necessary for optimal performance.
> >
> > Despite the appearance of complexity, the training settings for the first and third stages are essentially the same. The actual process involves inserting a training phase with a different strategy for the binary layer in the middle. The total training duration is not excessively long, encompassing only 110 epochs, with the relatively unstable second stage requiring just 10 epochs. The detailed settings of the three-stage training are presented in the following table.
> >
> > ---
> > **Training Parameter settings.**
> >
> > | config            | Stage 1-value              | Stage 2-value             | Stage3-value              |
> > | ----------------- | -------------------- | ------------------- | ------------------- |
> > | optimizer         | AdamW                | AdamW               | AdamW
> > | optimizer momentum($\beta_1,\beta_2$)|0.9, 0.999| 0.6,0.9999  |0.9,0.999
> > | base learning rate| 5e-4                 | 1e-4               |1e-4
> > | weight decay| 0.03                 | 5e-5               |0.03
> > | batch size| 4096                 | 4096               |4096
> > | learning rate schedule| cosine decay        | cosine decay      |cosine decay
> > | warmup epochs| 5                 | 1               |5
> > | frozen layers|none|mask|bi-conv-kernels,mask
> > | label smoothing| 0.1                 | 0.1               |0.1
> > | random erase| 0.25                 | 0               |0.25
> > | mixup| 0.8                 | 0               |0.8
> > | cutmix| 1.0                 | 0               |1.0
> > | drop path| 0.1                 | 0.1               |0.1
> >
> >
> > ---
> >
> > For a comprehensive understanding, we have provided detailed explanations of the three-stage training process in Section E of the Appendix. This should facilitate the replication of our approach by other researchers.
> >
> > ---
> >
> > > **W-4: The writing should be further improved, especially the mathematical notations in Section 3.3. The formula for binary compression is not formal.**
> >
> > ---
> > **REPLY**:We thank you for highlighting the need for improvement in the writing, particularly regarding the mathematical notations in Section 3.3. We have thoroughly reviewed the manuscript, focusing on refining any verbose, confusing, or ambiguous expressions. Special attention was given to improving Figure 1 to enhance its clarity and intuitiveness.
> >
> > In Section 3.3, we have revised the explanation of the binarization process. This includes a clearer presentation of the step function and a more rigorous definition of the binarization of the weight matrix $\boldsymbol{W}$, denoted as $\boldsymbol{W}^b$. We have clarified the relationship between $\boldsymbol{W}^b$ and $\boldsymbol{\Theta}$ and refined the calculation formula for s to be more precise. Additionally, we have included the formula for computing $\boldsymbol{\theta}$ during binarization.
> >
> > We believe these modifications have made the content in this section more rigorous and easier to understand, addressing your concerns effectively.

---

> > > ### Author Response · Authors · 2023-11-20
> > >
> > > ---
> > >
> > > > **W-5: The experiments are not sufficient and many critical comparisons are missing. For example, the backbone is ViT variants. However, the ViT is very computationally expensive because its computational complexity is quadratic to the input spatial size. This also embeds the real-time applications of HSI processing. In contrast, MST or MST++ are specially designed for HSI processing. They treat spectral feature maps as a token to capture the interdependencies between spectra with different wavelengths. Most importantly, they are very efficient with linear computational complexity regarding spatial resolution. Thus, it is better to add a comparison with the spectral Transformer in Figure 5.**
> > >
> > > ---
> > > **REPLY**:We greatly appreciate your recommendation of the MST and MST++ works. These studies indeed offer significant insights that can aid our research.
> > >
> > > Improving the efficiency of signal acquisition is a key aspect of enhancing the real-time capabilities of hyperspectral imaging systems. In our work, the DMD signal acquisition system's primary time-consuming factor is the signal acquisition process. Therefore, our focus has been on enhancing signal acquisition efficiency, which directly translates to achieving lower undersampling rates in our experiments.
> > >
> > > While MST and MST++ are commendable for their efficiency in processing hyperspectral data, their focus is primarily on data processing efficiency. Our experiments, however, are centered on evaluating performance under low undersampling rates. This difference in focus means that a direct comparison with MST and MST++ might not fully align with the specific objectives of our study. In Figure 5, we have chosen to compare methods that utilize learnable masks on patches, as with LUM-ViT, to more effectively validate our mask strategy. We acknowledge the significance of MST and MST++ in their respective areas, although their designs differ from the scope of our experiments.
> > >
> > > Nevertheless, we recognize the importance of these works and the role they play in advancing data processing speed, a key direction for improving system real-time capabilities. We have introduced MST and MST++ in the introduction of our manuscript as significant references for future exploration.
> > >
> > > Regarding the concern about the sufficiency of experiments, we have added experiments with Random-mask-ViT and Mag-mask-ViT as baseline methods to validate the effectiveness of our learnable mask strategy. These results are presented in Figure 4 and detailed in Section B of the Appendix. Additionally, we have included classification experiments on three hyperspectral remote sensing datasets to further validate LUM-ViT's capabilities in processing actual hyperspectral data. These experiments are presented in the newly added subsection 'The Hyperspectral Image Classification Experiments' and detailed in Section A of the Appendix.
> > >
> > > ---
> > >
> > > > **W-6: The binarization mechanism is out of fashion. BiSCI provides a specially designed binarized convolution unit BiSR-conv block to process HSI data cubes. It is also better to add a comparison with this new technique.**
> > >
> > > ---
> > > **REPLY**:We are grateful for your suggestion regarding the BiSCI work, which offers innovative approaches to the binarization process in models. The use of the Tanh function in this work is particularly inspiring and provides valuable insights for our research.
> > >
> > > We recognize that there have been significant advancements in binarization strategies recently, indicating substantial progress in this field.
> > >
> > > Our method primarily focuses on the design and experimental validation of the learnable mask. Given that DMD can only perform binary on/off modulation, we employed binarization techniques to adapt to DMD-based computing. Binarization, while not the central focus of our approach, was implemented using a basic binarization method due to this requirement.
> > >
> > > We understand that improved binarization techniques can enhance training stability. In future work, exploring more effective binarization strategies will be a priority for us. The binarization strategy used in BiSCI, particularly intriguing to us, is mentioned in the subsection 'Kernel-level weight binarization'. We see great potential in incorporating such innovative techniques into our future research to further improve our model's performance.

---

> > > > ### Author Response · Authors · 2023-11-20
> > > >
> > > > ---
> > > >
> > > > > **Q-1: The technical route and core idea in this paper is to accelerate the HSI processing, which is similar to Coded Aperture Spectral Snapshot Imaging (CASSI). Could you please analyze the differences, advantages, and disadvantages of these two systems?**
> > > >
> > > > ---
> > > > **REPLY**:Your question about the similarities and differences between our approach and Coded Aperture Spectral Snapshot Imaging (CASSI) is intriguing. The primary distinction lies in the hardware systems used. Our system employs a DMD signal acquisition system, which is detailed in Section D of the Appendix. This system captures signals using a spectrometer with high spectral resolution, in our case, 2048 spectral channels. However, a spectrometer can only capture single-point pixel information at a time, requiring the DMD to reflect different areas of the target object to the spectrometer with various masks. This process, if done point-by-point, is time-consuming. Originally, this system was mainly used in single-pixel imaging based on compressed sensing principles, to reduce the number of required samplings.
> > > >
> > > > In contrast, the CASSI system superimposes information from multiple spectral channels onto a single grayscale image using a disperser and reconstructs the original image's spectral channels algorithmically based on a random mask applied before dispersion.
> > > >
> > > > Both systems aim to create a three-dimensional hyperspectral cube. The DMD system has a fixed spectral resolution, while CASSI has a fixed spatial resolution. Both require algorithmic optimization to acquire their respective to-be-capture dimensions. Deep learning methods can significantly contribute to this process.
> > > >
> > > > When using compressed sensing algorithms in the DMD system, random masks displayed on the DMD are common for signal acquisition, followed by algorithmic reconstruction. This shares similarities with the random mask usage in CASSI. In our work, we replace the random mask in the DMD system with network parameters trained through deep learning, and we speculate that this approach might also have potential in the CASSI system.
> > > >
> > > > Cost-wise, the primary expense in the DMD system is the DMD itself, which may be more economical compared to the CASSI system.
> > > >
> > > >
> > > > We hope this response has addressed your question and clarified the differences, advantages, and potential of these two systems.

---

### Official Review · Reviewer_vQM3 · 2023-10-31

**Soundness:** 3 good
**Presentation:** 3 good
**Contribution:** 3 good
**Rating:** 6
**Confidence:** 4

**Summary:**

This paper proposes a novel approach using pre-acquisition modulation with a deep learning model called LUM-ViT. Specifically, it utilizes ViT as the backbone network and a DMD signal acquisition system for patch-embedding. Moreover, a kernel-level weight binarization technique and a three-stage fine-tuning strategy is proposed for optimizing the optical calculations. With low sampling rates, LUM-ViT maintains high accuracy on ImageNet dataset.

**Strengths:**

1.	The idea is of the paper is interesting. The proposed method performs calculations of the patch-embedding layer instead of directly sampling the whole images. The proposed LUM-ViT is suited for both dataset and downstream tasks.
2.	The accuracy loss is low with extreme under-sampling

**Weaknesses:**

1.	The description of the entire system and methods is not clear and intuitive enough, and the figures are also misleading. The RGB image is used as an example in Figure 1, which does not reflect the characteristics of hyperspectral imaging.
2.	Lack of experiments on real hyperspectral imaging. The author has emphasized hyperspectral imaging in the introduction section, but in reality, it has not been verified using real hyperspectral images. The performance of this method in real hyperspectral imaging tasks still needs to be discussed

**Questions:**

1.	The training phase uses images of 3 color channels while the real-world experiment uses 7 color images. What is the meaning of ‘reconfigured’? Or the author just fine tuned the LUM-VIT with 7 color samples? Can the pre trained model be used directly for images in different bands without the need for matching data?
2.	This acquisition system seems to have to work together with vit to obtain intermediate features of images. Can it reconstruct a complete hyperspectral image in a real-world environment?

---

> ### Author Response · Authors · 2023-11-19
> **Reply to reviewer vQM3**
>
> ## Reply to reviewer vQM3
>
> Dear Reviewer,
>
> We sincerely thank you for your constructive suggestions, which have been beneficial in enhancing our research. Your interest in the paper's idea and recognition of the LUM-ViT model's performance are greatly appreciated and serve as valuable encouragement to our team.
>
> We apologize for the delay in our response. This was due to conducting comprehensive experiments on three hyperspectral datasets, as suggested in your review. These crucial experiments aimed to address the weaknesses you pointed out and to strengthen the validation of our method in real hyperspectral imaging scenarios.
>
> For ease of review, all modifications made in the manuscript are marked in red, except for the content in the Appendix, which is entirely new.
>
> We will now address each of the weaknesses and questions you raised in detail.

---

> > ### Author Response · Authors · 2023-11-19
> >
> > ---
> >
> > > **W-1: The description of the entire system and methods is not clear and intuitive enough, and the figures are also misleading. The RGB image is used as an example in Figure 1, which does not reflect the characteristics of hyperspectral imaging.**
> >
> > ---
> > **REPLY**:We acknowledge that the description of our method could have been clearer, and we appreciate your feedback highlighting this issue. We suspect the lack of clarity stemmed primarily from the insufficiently distinct introduction of the two main stages of our method: the Training Phase and the Real-World Application Phase. The Training Phase involves training and inference on electronic computing hardware, while the Real-World Application Phase involves real-world performance testing with the DMD signal acquisition system. These two phases are sequential.
> >
> > To address this, we have made the following modifications:
> >
> > 1. **Terminology**: We have capitalized the first letters of “the Training Phase” and “the Real-World Application Phase” throughout the manuscript to signify their specific importance.
> >
> > 2. **Preliminaries Subsection**: We have added detailed descriptions of these two phases at the beginning of the "Preliminaries" subsection to help readers understand the distinction and sequence clearly.
> >
> > 3. **Figure 1 Revision**: We realized that the original differentiation between the two phases in Figure 1, marked by grey and blue colors, was not intuitive due to the low visibility of grey. We have now changed this to green and blue for clearer distinction and added annotations directly in the figure, in addition to the caption, to explain the meaning of these colors.
> >
> > We also acknowledge that the initial lack of detail about the DMD hardware could have contributed to the confusion. To remedy this, we have included a comprehensive description of DMD in Section D of the Appendix.
> >
> > Regarding the use of an RGB image in Figure 1 and its potential for misrepresentation, we have made revisions. We now overlay the original image with its color maps under different spectral masks to represent the inclusion of information across numerous spectral channels. This modification clarifies that the image is indeed hyperspectral, with annotations added directly on the figure for emphasis.
> >
> > We hope these modifications adequately address the weaknesses you identified and improve the overall clarity and accuracy of our method's presentation.
> >
> >
> > ---
> >
> > > **W-2: Lack of experiments on real hyperspectral imaging. The author has emphasized hyperspectral imaging in the introduction section, but in reality, it has not been verified using real hyperspectral images. The performance of this method in real hyperspectral imaging tasks still needs to be discussed**
> >
> > ---
> > **REPLY**:Thank you for your insights regarding the scope of our experimental validation. We appreciate your point on the necessity of a broader range of tests to comprehensively assess the performance of our model, LUM-ViT, particularly in the context of hyperspectral imagery.
> >
> > ---
> >
> > **Results on HSI classification.**
> >
> > | Dataset           | OA (%) - before mask | OA (%) - after mask |
> > | ----------------- | -------------------- | ------------------- |
> > | Indian Pines      | 87.4                 | 86.2                |
> > | Pavia University  | 89.5                 | 88.6                |
> > | Salinas           | 99.4                 | 99.1                |
> >
> > ---
> >
> > Recognizing the importance of diversifying our validation scenarios, we have conducted additional experiments on three widely-used hyperspectral remote sensing datasets: Indian Pines, Pavia University, and Salinas. These datasets are commonly utilized for hyperspectral pixel and region classification tasks, and we have tested LUM-ViT on these tasks. The results on these three datasets were promising, demonstrating LUM-ViT's capability in processing actual spectral data effectively.
> >
> > We have added a new subsection titled 'The Hyperspectral Image Classification Experiments' in the 'Experiments' section of our manuscript to report the results of LUM-ViT on these three hyperspectral datasets. Due to space constraints, details regarding the datasets and parameter adjustments for these experiments have been included in Section A of the Appendix.
> >
> > Once again, thank you for your insightful feedback, which has prompted us to further strengthen our manuscript with these additional experiments.

---

> > > ### Author Response · Authors · 2023-11-19
> > >
> > > ---
> > >
> > > > **Q-1: The training phase uses images of 3 color channels while the real-world experiment uses 7 color images. What is the meaning of ‘reconfigured’? Or the author just fine tuned the LUM-VIT with 7 color samples? Can the pre trained model be used directly for images in different bands without the need for matching data?**
> > >
> > > ---
> > > **REPLY**:Regarding your question about the training phase using 3-color channel images and the real-world experiment using 7-color images, 'reconfigured' refers to the adjustment of spectral weighting parameters $\boldsymbol{V}$ based on spectral data. This data can be simulated by measuring the spectra of seven colors displayed on an ink screen, akin to how RGB images would appear on the screen. The spectral data is what we have presented in Figure 6 (a).
> > >
> > > As the ImageNet images used do not include extensive spectral channel information, this adjustment process is relatively straightforward. On this basis, further fine-tuning with a small set of samples allows us to achieve satisfactory accuracy.
> > >
> > > It is challenging to directly apply the pre-trained model to real-world tasks due to hardware limitations. The hardware system introduces noise and optical biases, some of which are caused by detection instruments, while others result from insufficient optical system correction. To make the pre-trained model robust against these errors, it is necessary to design artificial errors during the training process for targeted training against these specific types of inaccuracies.
> > >
> > > We hope these clarifications and methodological adjustments effectively address the concerns raised in your question.
> > >
> > >
> > >
> > > ---
> > >
> > > > **Q-2: This acquisition system seems to have to work together with vit to obtain intermediate features of images. Can it reconstruct a complete hyperspectral image in a real-world environment?**
> > >
> > > ---
> > > **REPLY**:This is an intriguing question indeed. Theoretically, it's possible to achieve reconstruction of the input data by modifying the downstream task, as the requirements for classification and reconstruction tasks are similar in terms of model demands. The success of self-supervised learning in pre-training tasks, which perform well, supports this possibility.
> > >
> > > However, we suspect that the quality of reconstruction may not be satisfactory under stringent under-sampling conditions. The reason LUM-ViT achieves satisfactory results in classification tasks at low undersampling rates is due to the sparsity of information required for these tasks in image or hyperspectral data. The amount of information needed for classification is significantly less than the total information in the original image, which makes the feature extraction process feasible. In contrast, reconstruction tasks demand a larger amount of information. At low undersampling rates, the information captured by the model may not be sufficient to accurately reconstruct the complete input image.
> > >
> > > Despite these challenges, we still believe that pursuing this direction for reconstruction is viable and investigating this reconstruction process is an intriguing aspect of future research.

---

### Official Review · Reviewer_rvtr · 2023-11-01

**Soundness:** 3 good
**Presentation:** 3 good
**Contribution:** 3 good
**Rating:** 6
**Confidence:** 3

**Summary:**

This paper proposes a learnable under-sampling mask vision Transformer, which incorporates a learnable undersampling mask tailored for pre-acquisition modulation.

**Strengths:**

+ The paper is well-organized and clearly written.

+ The proposed three-stage training strategy for training LUM-ViT is effective.

**Weaknesses:**

- Technical details should be clear. How to achieve the learnable under-sampling mask? How is this learnable achieved? Is the learning accurate? Relevant visualization results should be provided.

- The experimental results seem insufficient. The author only conducted validation on the ImageNet-1k classification task, and other tasks should also be further explored.

-----------------------After Rebuttal---------------------------

Thank you for your feedback. The rebuttal addressed my concerns well. Considering other reviews, I have decided to increase my score.

**Questions:**

See the above Weaknesses part.

---

> ### Author Response · Authors · 2023-11-19
> **Reply to reviewer rvtr**
>
> ## Reply to reviewer rvtr
>
> Dear Reviewer,
>
> First and foremost, we would like to express our sincere gratitude for your thoughtful and constructive feedback on our paper. Your positive remarks about the organization and clarity of our manuscript, as well as the effectiveness of the proposed three-stage training strategy for training LUM-ViT, are greatly appreciated. Such recognition from an expert in the field is both encouraging and affirming for our work.
>
> We apologize for the delay in our response. The additional time was utilized to conduct extensive experiments with two comparative methods and on three hyperspectral datasets. These experiments, while time-consuming, were essential in addressing the weaknesses you highlighted. We believe that the results of these additional experiments have significantly strengthened our paper and have allowed us to thoroughly address your concerns.
>
> To ensure clarity and ease of review, we have marked all modifications made in the manuscript in red for easy identification, with the exception of the content in the Appendix, as it is newly added. These changes include adjustments and additions in response to valuable feedback, as well as results from additional experiments.
>
> In the following sections of this response, we will specifically address each weakness you have identified, providing detailed explanations and supplementary information.
>
> Thank you once again for your valuable insights and for the opportunity to enhance the quality of our work.

---

> ### Author Response · Authors · 2023-11-19
>
> ---
>
> > **W-1: Technical details should be clear. How to achieve the learnable under-sampling mask? How is this learnable achieved? Is the learning accurate? Relevant visualization results should be provided.**
>
> ---
>
> **REPLY**: Thank you for highlighting the need for clearer technical details regarding the learnable under-sampling mask in our LUM-ViT model. We acknowledge that our initial manuscript may not have adequately explained the implementation of the learnable mask.
>
> To clarify, the learnable under-sampling mask in LUM-ViT is achieved through a binary decision mask with trainable parameters. During the training phase, these parameters determine the mask values, which influence the loss function values of two critical metrics: the accuracy of downstream tasks and the under-sampling rate. These parameters are updated through backpropagation, resulting in a mask that adheres to the under-sampling rate criteria and is optimally adapted for the downstream tasks.
>
> It is important to note that, unlike DynamicViT or AdaViT architecture, which dynamically adjust their masks during inference, LUM-ViT requires pre-acquisition determination of the mask scheme. LUM-ViT focuses on under-sampling, which means reducing the number of optical computations and detections. For a single inference, optical computations are carried out before the entire system detects the target object and cannot adjust the masking strategy based on yet-to-know information on the target object. Consequently, LUM-ViT generates a mask that remains fixed post-training, *i.e.*, during the inference stage. This mask is designed to suit the validation set and downstream tasks through training on a compatibly distributed training set.
>
> To further elucidate the principle of the learnable mask, we have revised our manuscript to replace the term 'learnable parameters' with 'trainable parameters'. This change emphasizes that the learnability is achieved through training. Additionally, we have substantially revised the subsection 'Learnable Under-sampling Mask' in the manuscript and revised the description in the introduction regarding the different requirements for learnable masks in under-sampling tasks versus those in pruning tasks.
>
> ---
>
> **Top-1 Acc comparison.**
>
> | Model             | Acc (%) - 10% mask | Acc (%) - 2% mask |
> | ----------------- | ------------------------ | ----------------------- |
> | Random-mask-ViT   | 74.6                     | 62.1                    |
> | Mag-mask-ViT      | 76.2                     | 65.4                    |
> | **LUM-ViT**       | **81.9**                 | **78.3**                |
>
> ---
>
> We have also incorporated new experiments to validate the efficacy of the learnable mask approach. We introduced two alternative mask schemes: Random-mask-ViT, which employs a random mask, and Mag-mask-ViT, which uses a mask based on data magnitude. The results of these experiments, which show a significant performance gap between these methods and LUM-ViT, indirectly corroborate the effectiveness of our learnable mask method.
>
> These experiments are detailed in the subsection 'The Training Phase Experiments', with results showcased in Fig. 4. Due to space constraints, we have included detailed descriptions of the Random-mask-ViT and Mag-mask-ViT models, along with previously base models like DU-ViT and CS-method, in Section B of the Appendix.
>
> Furthermore, we have added a visualization analysis to investigate the internal mechanism of the learnable mask. Since LUM-ViT requires the mask to be fixed before sample detection during inference and does not allow for dynamic adjustment based on individual samples, we are unable to provide visualizations that compellingly illustrate non-essential parts of an individual sample being selectively masked. To compensate for this, we have visualized the masking results of LUM-ViT on convolutional kernels and patches at different mask ratios, using heatmaps and histograms. This analysis reveals LUM-ViT's tendency to select convolutional kernels under low under-sampling pressure and delve deeper into patch selection under high pressure. Due to space constraints, only the heatmap for LUM-ViT-2% is retained in the main text, with a detailed analysis provided in Section C of the Appendix.
>
> Once again, we thank you for your insightful comments, which have significantly contributed to the improvement of our manuscript.

---

> ### Author Response · Authors · 2023-11-19
>
> ---
>
> > **W-2: The experimental results seem insufficient. The author only conducted validation on the ImageNet-1k classification task, and other tasks should also be further explored.**
>
> ---
>
> **REPLY**: Thank you for your insights regarding the scope of our experimental validation. We appreciate your point on the necessity of a broader range of tests to comprehensively assess the performance of our model, LUM-ViT, particularly in the context of hyperspectral imagery.
>
> ---
>
> **Results on HSI classification.**
>
> | Dataset           | OA (%) - before mask | OA (%) - after mask |
> | ----------------- | ---------------------------------- | --------------------------------- |
> | Indian Pines      | 87.4                               | 86.2                              |
> | Pavia University  | 89.5                               | 88.6                              |
> | Salinas           | 99.4                               | 99.1                              |
>
> ---
>
> Recognizing the importance of diversifying our validation scenarios, we have conducted additional experiments on three widely-used hyperspectral remote sensing datasets: Indian Pines, Pavia University, and Salinas. These datasets are commonly utilized for hyperspectral pixel and region classification tasks, and we have tested LUM-ViT on these tasks. The results on these three datasets were promising, demonstrating LUM-ViT's capability in processing actual spectral data effectively.
>
> We have added a new subsection titled 'The Hyperspectral Image Classification Experiments' in the 'Experiments' section of our manuscript to report the results of LUM-ViT on these three hyperspectral datasets. Due to space constraints, details regarding the datasets and parameter adjustments for these experiments have been included in Section A of the Appendix.
>
> Once again, thank you for your insightful feedback, which has prompted us to further strengthen our manuscript with these additional experiments.

---

> ### Author Response · Authors · 2023-11-22
>
> Dear Reviewer,
>
> I am writing in reference to manuscript ID 9181. We have previously addressed the feedback provided and are keen to hear any additional thoughts or comments you might have. Your insights are invaluable to the continued refinement of our work.
>
> We have diligently worked on improving the manuscript, with significant enhancements in clarity and detail, especially in the Appendix, where we have elaborated on the experimental procedures. The main code for the LUM-ViT model is also available now, aiming to support the reproducibility of our findings.
>
> Understanding the many demands on your time, we appreciate any further input you can provide at your earliest convenience. As the review process deadline nears, your further guidance will be instrumental in ensuring the quality and accuracy of our research.
>
> Thank you very much for your attention and contribution to our work. We look forward to your valuable feedback.
>
> Warm regards,
> Manuscript ID 9181

---

### Meta-Review · Area_Chair_gVkm · 2023-12-10

**Metareview:**

The authors propose LUM-ViT, a learnable under-sampling mask vision transformer for bandwidth-limited optical signal acquisition.
LUM-ViT employs ViT and prior information to optimize for optical calculations and reduce acquisition volume. A DMD signal acquisition system is employed for patch-embedding, a kernel-level weight binarization technique and a three-stage fine-tuning strategy are proposed for optimizing the optical calculations. The experiments demonstrate the advantages of the proposed LUM-ViT and its practicality.

The major strengths are the studied problem, the idea and the novel ViT-based solution, the performance and the practicality.

The major weaknesses identified by the reviewers were: not reporting on real datasets, missing details for reproducibility, unclear descriptions.

The authors provided responses to all the reviewers' concerns, new results, an updated manuscript and details and engagement to release the codes for reproducibility and the reviewers are now unanimously leaning towards acceptance of the paper (6,6,6).

The meta-reviewer after carefully reading the reviews, the discussions, and the paper, agrees with the reviewers and recommends acceptance.

**Justification For Why Not Higher Score:**

While the studied problem, the novel solution and the performance are strengths, the advances at the machine learning or theoretical level are limited and the interest for the addressed topic is rather limited within the ICLR community.

**Justification For Why Not Lower Score:**

Three reviewers and the meta-reviewer agree that the paper has merits, and the contributions are sufficient and of interest. There is no significant flaw to impede publication.

---

### Decision · Program_Chairs · 2024-01-16

Accept (poster)